# Effects of *Eucalyptus* Biochar on Intestinal Health and Function in Largemouth Bass (*Micropterus salmoides*)

**DOI:** 10.3390/biology14121754

**Published:** 2025-12-07

**Authors:** Bing Fu, Yan Chen, Xiang Li, Huiyun Zhou, Junru Hu, Jinghong Li, Wen Huang, Hongxia Zhao, Bing Chen, Jiun-Yan Loh

**Affiliations:** 1Yazhou Bay Innovation Institute, Hainan Tropical Ocean University, Sanya 527022, China; 15869704640@163.com (B.F.); cy969398hn@126.com (Y.C.); 2Institute of Animal Science, Guangdong Academy of Agricultural Sciences/Aquatic Research Center, Guangdong Academy of Agricultural Sciences/Key Laboratory of Animal Nutrition and Feed Science in South China, Ministry of Agriculture and Rural Affairs/Guangdong Key Laboratory of Animal Breeding and Nutrition, Guangzhou 510640, China; hujunru1025@163.com (J.H.); morepapers123@126.com (J.L.); zhaohongxia8866@163.com (H.Z.); 3Sericultural & Agri-Food Research Institute, Guangdong Academy of Agricultural Sciences/Key Laboratory of Functional Foods, Ministry of Agriculture and Rural Affairs/Guangdong Key Laboratory of Agricultural Products Processing, Guangzhou 510610, China; 4Institute of Agricultural Resources and Environment, Guangdong Academy of Agricultural Sciences, Guangzhou 510640, China; lixiang142213@163.com; 5TC and JN Limited Liability Company, Chula Vista, CA 91913, USA; jimchow628@gmail.com; 6Tropical Futures Institute, James Cook University, Sims Drive, Singapore 387380, Singapore

**Keywords:** biochar, largemouth bass, intestinal health, inflammation, intestinal microbiota, metabolomics

## Abstract

This study investigates the effects of Eucalyptus-derived biochar as a natural feed additive on the intestinal health of largemouth bass (*Micropterus salmoides*). The findings show that while biochar supplementation did not significantly alter intestinal tissue morphology. Results from 16S rDNA sequencing further revealed that biochar increased the relative abundance of Firmicutes and *Lactococcus* while reducing harmful bacterial populations. Similarly, biochar improved overall health status by promoting isoflavone biosynthesis and bile acid and amino acid metabolism, and by lowering intestinal absorption of soyasaponins under aquaculture conditions. These results suggest that biochar is a valuable dietary supplement for modulating inflammatory responses, strengthening the intestinal barrier, and supporting the health of farmed largemouth bass.

## 1. Introduction

In recent years, global population growth, changing consumer preferences, and a rising emphasis on nutritional health have steadily increased the demand for aquatic products. As a result, aquaculture systems have become progressively more intensive [1,2]. For largemouth bass (*Micropterus salmoides*), an economically important freshwater species, intestinal integrity is essential for nutrient absorption, immune defense, and overall growth performance. Under high-density farming conditions, factors such as oxidized dietary lipids, mycotoxins, and waterborne pollutants, including ammonia nitrogen and nitrite, can induce oxidative stress, damage the intestinal mucosa, disrupt microbial homeostasis, and trigger inflammatory responses. These disturbances compromise fish health and diminish the profitability of *M. salmoides* culture [3,4,5,6,7]. Therefore, the development of safe and effective feed additives has become a priority in aquatic nutrition research to protect intestinal health in this species.

Biochar, a porous, carbon-rich material produced by biomass pyrolysis, has a high specific surface area, diverse functional groups, and strong adsorption capacity. These features underpin its broad potential applications in soil enhancement, environmental remediation, and livestock production [8,9]. *Eucalyptus* species, commonly cultivated for timber, generate substantial quantities of branch residues that are often discarded. When converted to biochar through pyrolysis, these residues can be repurposed for various uses, including wastewater treatment and improving intestinal health in animals. The unique physicochemical properties of *Eucalyptus* biochar, such as its high porosity, alkaline nature, and mineral content, contribute to these benefits [10,11,12,13].

Previous studies indicate that biochar can support the growth and health of aquatic animals by adsorbing harmful substances, improving water quality, modulating the gut microbiota, and enhancing immune responses [8,14]. However, most research has focused primarily on physiological, biochemical, and morphological parameters. To date, the effects of *Eucalyptus* branch biochar on the intestinal health of aquatic species, especially *M. salmoides*, remain insufficiently explored, and the underlying regulatory mechanisms are poorly understood.

This study provides a comprehensive assessment of the influence of *Eucalyptus* biochar on intestinal morphology, digestive enzyme activity, inflammatory gene expression, gut microbial composition, and metabolomic profiles in *M. salmoides*. The findings aim to clarify how *Eucalyptus* biochar promotes intestinal health, facilitates the high-value utilization of *Eucalyptus* by-products, and contributes to the development of eco-friendly feed additives for sustainable aquaculture.

## 2. Materials and Methods

### 2.1. Experimental Fish

Healthy *Micropterus salmoides* were sourced from Guangdong Heshi Aquatic Products Co., Ltd. (Foshan, China). All animal procedures were approved by Hainan Tropical Ocean University and the Institute of Animal Science, Guangdong Academy of Agricultural Sciences, and were conducted in accordance with relevant institutional and national guidelines. Before the experiment, fish were acclimated for one week in rectangular cement tanks (3.0 m × 2.0 m × 1.2 m) and fed the basal diet (G0, Table 1). Following acclimation, fish were counted, bulk-weighed, and determined to have an average initial body weight of 13.34 g. The feeding trial was then carried out in 200 L circular tanks (0.8 m diameter, 0.7 m height), stocked at 30 fish per tank, with three replicate tanks assigned to each dietary treatment.

### 2.2. Experimental Feeding and Diets

*Eucalyptus* biochar was provided by the Institute of Agricultural Resources and Environment, Guangdong Academy of Agricultural Sciences. Its preparation involved cleaning and drying eucalyptus branches (approximately 10 cm in length), followed by pyrolysis in a muffle furnace at 500 °C with a heating rate of 10 °C min^−1^ for 2 h. The resulting material was crushed, passed through a 60-mesh sieve, acidified with 0.5% hydrochloric acid for 72 h, rinsed with clean water to a neutral pH, dried at 60 °C, and stored at 4 °C. The physicochemical characteristics of the biochar are presented in Appendix A. Six experimental diets were formulated with 0, 2.5, 5.0, 10.0, 20.0, and 40.0 g kg^−1^ biochar, designated G0 through G5, respectively (Table 1). These inclusion levels were selected based on previous research in gilthead seabream (*Sparus aurata*) [15] and giant trevally (*Caranx ignobilis*) [16]. All feed ingredients were ground to pass through a 60-mesh sieve, thoroughly mixed, and moistened with 30% water. The mixture was extruded using a T-50 twin-screw extruder (Huaqiang Extrusion Machinery Factory, Guangzhou, China) at 115 ± 5 °C to produce 3.0 mm pellets. Pellets were then coated with oil, dried at 54 °C for 6 h, and stored at 4 °C until use.

Throughout the feeding trial, fish were fed to apparent satiation twice daily at 09:00 and 17:00. One-third of the tank water was replaced daily with dechlorinated water. Water quality parameters were maintained at 28.0 ± 2.0 °C for temperature, 5.50 ± 0.40 mg L^−1^ for dissolved oxygen, 7.75 ± 0.25 for pH, and 0.16 ± 0.03 mg L^−1^ for NH_4_-N.

### 2.3. Fish Intestinal Sampling

At the end of the feeding trial, fish were fasted for 24 h and euthanized by immersion in 40 mg L^−1^ MS-222 (Sigma, St. Louis, MO, USA). After dissection on ice, intestinal samples were collected from nine fish per tank across the three replicate tanks in each treatment group. Entire intestines from three fish were stored at −80 °C for digestive enzyme assays, and another three were preserved at −80 °C for mRNA expression analysis. Intestinal contents from an additional three fish were collected and frozen at −80 °C for 16S rDNA sequencing and metabolomic analyses. The foregut of the remaining three fish was fixed in 4% paraformaldehyde for histological examination. Morphometric assessments, digestive enzyme activity measurements, and mRNA expression analyses were conducted using individual fish as independent biological replicates (*n* = 9 per group: 3 tanks × 3 fish). Metabolomic and microbiota analyses were performed on composite samples generated by pooling intestinal tissues or contents from three fish within each tank (one composite sample per tank, *n* = 3).

### 2.4. Intestinal Morphology Analysis

Foregut segments (1 cm) were fixed in 4% paraformaldehyde for 24 h, embedded in paraffin, dewaxed with xylene, and rehydrated through a graded ethanol series. Tissue sections (4 μm) were stained with hematoxylin and eosin (H&E) following standard procedures. Images were obtained using an optical microscope (PANNORAMIC DESK/MIDI/250/1000, 3DHISTECH, Budapest, Hungary) and a photographic microscope (Eclipse Ci-L, Nikon, Tokyo, Japan). Morphometric parameters, including villus length and width, goblet cell number, epithelial length, and muscular layer thickness, were quantified using Image-Pro Plus 6 (Media Cybernetics, Rockville, MD, USA) as previously described [17].

### 2.5. Digestive Enzymes Analysis

Approximately 0.1 g of intestinal tissue was homogenized in 0.9% saline at a 1:9 (*w*/*v*) ratio on ice. The homogenate was then centrifuged at 2500× *g* for 10 min at 4 °C, and the resulting supernatant was collected. Activities of trypsin (No. A080-2), amylase (No. C016-1-1), and lipase (No. A054-2-1) were determined using commercial assay kits (Nanjing Jiancheng Bioengineering Institute, Nanjing, China), following the procedure described by Wang et al. [18].

### 2.6. Messenger Ribonucleic Acid (mRNA) Expression Analysis

Total RNA was extracted from approximately 50 mg of intestinal tissue using TRIzol reagent (Hubei Chucheng Zhengmao Technology Engineering Co., Ltd., Wuhan, China). RNA concentration and integrity were assessed by 1% agarose gel electrophoresis (Bio-Rad, Hercules, CA, USA). cDNA was synthesized using the HiScript RT SuperMix for qPCR kit (Vazyme, Nanjing, China). Quantitative PCR was conducted in a 20 μL reaction mixture containing 1.0 μL cDNA, 10.0 μL of 2× Taq Pro Universal SYBR qPCR Master Mix, 0.8 μL of each primer (10 μM), and 8.2 μL ddH_2_O. The amplification protocol consisted of an initial denaturation at 95 °C for 30 s, followed by 40 cycles of 95 °C for 10 s and 60 °C for 30 s. β-actin was used as the reference gene, and relative expression levels were calculated using the 2^−ΔΔCT^ method [19]. Primer sequences are provided in Table 2.

### 2.7. Gut Microbiota Analysis

DNA from pooled intestinal content samples was extracted using the E.Z.N.A.^®^ Soil DNA Kit (Omega Bio-tek, Norcross, GA, USA). DNA concentration and purity were assessed with a NanoDrop 2000 spectrophotometer (Thermo Fisher Scientific, Waltham, MA, USA). Primers targeting the hypervariable V3–V4 regions of the 16S rDNA gene were used for PCR amplification. PCR products were purified using a PCR Clean-Up Kit (Yuhua, Nanjing, China) and quantified with a Qubit 4.0 fluorometer (Thermo Fisher Scientific, Waltham, MA, USA). Sequencing libraries were then constructed and sequenced on the Illumina PE250/PE300 platform (Majorbio, Shanghai, China). Operational taxonomic units (OTUs) were clustered at 97% similarity using UPARSE 7.1, and taxonomic assignments were performed by comparing the representative sequences with the Silva 16S rRNA gene database (v138) using the RDP classifier (version 2.11).

### 2.8. Intestinal Content Metabolomics Analysis

Intestinal tissues were collected for metabolomic profiling. Sample preparation, liquid chromatography–mass spectrometry (LC–MS) analysis, and metabolite identification were performed as described by Li et al. [20]. In brief, approximately 50 mg of pooled intestinal content samples were placed into microcentrifuge tubes and extracted with 500 μL of 80% methanol. The mixture was homogenized at −10 °C and 50 Hz for 6 min, followed by ultrasonic extraction at 5 °C and 40 kHz for 30 min. Samples were then kept at −20 °C for 30 min and centrifuged at 13,000× *g* for 15 min at 4 °C. A 180 μL aliquot of the supernatant was collected for LC–MS/MS analysis, and an additional 20 μL was pooled to create quality control samples.

Chromatographic separation was performed using an ACQUITY UPLC HSS T3 column (100 mm × 2.1 mm, 1.8 μm; Waters, Milford, MA, USA) at 40 °C with a 3.0 μL injection volume. The mobile phases consisted of: A, 95% water and 5% acetonitrile containing 0.1% formic acid; and B, 47.5% acetonitrile, 47.5% isopropanol, and 5% water containing 0.1% formic acid. Mass spectrometry was conducted with the following settings: scan range *m*/*z* 70–1050; heater temperature 425 °C; spray voltage ± 3.5 kV; S-Lens RF level 50; sheath gas flow 50 arb; auxiliary gas flow 13 arb; capillary temperature 325 °C; collision energy 20/40/60 eV; MS2 resolution 7500; and full MS resolution 60,000.

Raw data were processed using Progenesis QI software 2.0 (Waters Corporation, Milford, MA, USA), and metabolite identification was achieved by matching MS and MS/MS spectra against established databases such as HMDB (http://www.hmdb.ca/, accessed on 23 October 2025) and METLIN (https://metlin.scripps.edu/, accessed on 23 October 2025).

### 2.9. Statistical Analysis

Data were analyzed using SPSS 22.0. A one-way ANOVA followed by Duncan’s multiple-range test was used to evaluate differences among treatment groups, and results are expressed as mean ± standard error (SE). Statistical significance was set at *p* < 0.05. For gut microbiota and metabolomics analyses, comparisons between the control group (G0) and the G3 group were assessed using Student’s *t*-test. All figures were generated using GraphPad Prism 9.05 (GraphPad Software, San Diego, CA, USA). A statistical power analysis was conducted with G*Power 3.1 to assess sample size sensitivity and ensure robust statistical outcomes.

## 3. Results

### 3.1. Intestinal Morphology

As shown in Figure 1, dietary *Eucalyptus* biochar supplementation influenced the intestinal morphology of *M. salmoides*. Goblet cell numbers increased steadily with increasing biochar levels and were significantly higher in the G5 group than in the control (*p* < 0.05). As illustrated in Figure 2, the mucosal and muscular layers across all treatments remained intact and well-organized. The mucosal layer contained numerous well-developed villi, while the muscle layer displayed clear structural integrity with uniform staining. The lamina propria consisted of connective tissue with sparse lymphocytic infiltration.

### 3.2. Digestive Enzyme Activity

As shown in Figure 3, the activities of amylase, lipase, and trypsin did not differ significantly among the treatment groups (*p* > 0.05). However, amylase and lipase activities increased in groups G1–G4 compared with the control group (*p* > 0.05).

### 3.3. Gene Expression

#### 3.3.1. Tight Junction Protein-Associated Genes

As shown in Figure 4A, *Claudin-3* expression increased significantly in groups G1–G4 as biochar levels increased, compared with the control group (*p* < 0.05). In comparison, *ZO-1* expression did not differ significantly among the groups (*p* > 0.05).

#### 3.3.2. Inflammation-Related Genes

Dietary biochar significantly influenced the expression of inflammation-related genes (Figure 4B). Compared with the control group, *IL-10* expression was significantly upregulated in the G1, G2, G3, and G4 groups (*p* < 0.05). *IL-1β* expression was significantly downregulated in the G4 and G5 groups (*p* < 0.05), while *TNF-α* levels were reduced considerably in the G2 and G3 groups (*p* < 0.05). No significant differences in *TGF-β1* expression were observed among the groups (*p* > 0.05).

### 3.4. Gut Microbiota

#### 3.4.1. Diversity Analysis

Based on the expression patterns of tight junction and inflammation-related genes, the control and G3 groups were selected for gut microbiota and metabolomics analyses. The α-diversity indices of the intestinal microbiota are shown in Table 3, with sequencing coverage ranging from 99.90% to 99.95% in both groups. Compared with the control, the G3 group showed no significant differences in Ace, Chao1, Shannon, or Simpson indices (*p* > 0.05). Principal coordinate analysis (PCoA) was used to assess similarities and differences in microbial community structure. As shown in Figure 5A, the two groups demonstrated partial overlap. Figure 5B displays the distribution of unique and shared OTUs. A total of 629 OTUs were identified across all samples, with 128 OTUs (25.55%) shared between the groups. The G3 group had fewer OTUs than the control group (208 vs. 421).

#### 3.4.2. Composition of Gut Microbiota

As shown in Figure 5C,E, the G3 group showed a higher relative abundance of Firmicutes compared with the control group, whereas Proteobacteria and Fusobacteria showed decreased relative abundances (*p* > 0.05). At the genus level (Figure 5D,F), the relative abundances of *Citrobacter*, *Lactococcus*, *Candidatus Arthromitus*, and unclassified *Peptostreptococcaceae* increased in the G3 group (*p* > 0.05). At the same time, *Mycoplasma*, *Plesiomonas*, and *Klebsiella* decreased relative to the control (*p* > 0.05).

### 3.5. Metabolomics Analysis

Multivariate analysis revealed a clear separation between the G3 and control groups. This distinction was visible in the PCA score plot and further supported by the distinct clustering observed in the OPLS-DA model (Figure 6A,B). All sample points in both models fell within the 95% confidence interval. Permutation testing (Figure 6C) confirmed the validity of the OPLS-DA model, with R^2^Y values exceeding 90% in all groups, indicating strong explanatory power. However, the Q^2^ regression line intercept was greater than 0.5, suggesting a potential risk of overfitting in the OPLS-DA model.

Using a significance threshold of *p* < 0.05 in the *t*-test and VIP > 1 in the OPLS-DA model, differential metabolites (DMs) between the two groups were identified and ranked by fold change. Compared with the control group, the G3 group demonstrated significant upregulation of 42 metabolites and downregulation of 64 metabolites (Figure 7A). Metabolites such as baicalin, genistein, apigenin, taurochenodeoxycholate-7-sulfate, taurocholic acid 3-sulfate, taurochenodeoxycholate-3-sulfate, and arginylmethionine were significantly elevated in the G3 group (*p* < 0.05), whereas aerobactin, mycobactins, soyasaponin, and etoxazole were significantly reduced (*p* < 0.05) (Figure 7B).

The differential abundance score plot revealed that biochar supplementation significantly enhanced and upregulated the isoflavonoid biosynthesis pathway, while simultaneously downregulating the neutrophil extracellular trap (NET) formation pathway (Figure 8).

Spearman correlation analysis between dominant intestinal bacterial genera and DMs showed that apigenin, propofol glucuronide, arginylmethionine, and macluraflavanone were strongly negatively correlated with *Klebsiella* (*p* < 0.01). Furthermore, 2-anthraquinonesulfonic acid and chrysin-7-O-glucuronide displayed significant negative correlations with *Klebsiella* (*p* < 0.05). Soyasaponin I, soyasaponin II, and etoxazole were also significantly negatively correlated with *Lactococcus* (*p* < 0.05) (Figure 9).

## 4. Discussion

The structure of the intestine is closely linked to the integrity of the intestinal barrier, which is essential for maintaining homeostasis by facilitating nutrient absorption and defending against pathogenic invasion. In aquatic animals, increases in parameters such as muscle layer thickness, villus height and width, and goblet cell abundance are generally viewed as indicators of healthy intestinal development [21]. Previous studies have shown that biochar supplementation enhanced intestinal villus length and goblet cell numbers in common carp (*Cyprinus carpio*) [22], and activated charcoal increased villus height across the foregut, midgut, and hindgut of Nile tilapia (*Oreochromis niloticus*) in a dose-dependent manner [23]. In this study, the intestinal mucosal and muscular layers of *M. salmoides* remained intact, well-organized, and densely structured, with abundant villi across all treatment groups. Although biochar supplementation did not significantly alter villus height, villus width, or muscle layer thickness, a significant increase in goblet cell number was observed in the G5 group. Furthermore, as the supplementation of biochar increased, the length of epithelial cells initially rose and then declined. These results suggest that dietary biochar partially improves intestinal morphology in *M. salmoides*.

Digestive enzymes such as amylase, trypsin, and lipase are key indicators of an animal’s digestive capacity, and their activity levels are positively associated with dietary nutrient utilization efficiency [24]. Previous research has demonstrated that biochar supplementation enhances apparent feed digestibility in *Catla catla* [25], *Labeo rohita* [26], goats [27], and weaned piglets [28]. In European sea bass (*Dicentrarchus labrax*), intestinal amylase and lipase activity increased significantly when charcoal or activated carbon was added to the diet [29]. These effects may stem from (1) biochar’s high specific surface area, electrical conductivity, and abundance of surface functional groups, which enable adsorption of digestive enzyme inhibitors such as hydrogen sulfide (H_2_S) and ammonium (NH_4_^+^) [30]; and (2) its ability to enhance electron transfer during anaerobic fermentation, promoting acid-producing bacteria and boosting short-chain fatty acid (SCFA) production [31]. In this study, biochar supplementation did not significantly alter amylase, lipase, or trypsin activities in *M. salmoides*, although a general upward trend was noted with higher inclusion levels. This outcome may be due to measuring digestive enzyme activities across the entire intestinal tract, which likely diluted localized biochar-induced changes that occur primarily in major digestive and absorptive regions, such as the foregut.

This study showed that dietary biochar supplementation significantly reduced the intestinal expression of the pro-inflammatory cytokines *IL-1β* and *TNF-α*, while enhancing the expression of the anti-inflammatory cytokine IL-10 in *M. salmoides*. Increased *IL-10* expression may help suppress inflammatory responses by counterbalancing the effects of *IL-1β* and *TNF-α* [32]. Similar anti-inflammatory effects have been observed in other species: activated charcoal reduced *IL-1β* levels in the ileum and jejunum of broilers [33]. Similarly, Yıldızlı et al. [34] reported a linear decrease in *TNF-α* and *IL-6* expression in macrophages exposed to olive and apricot kernel biochar. These findings align with the well-recognized hemostatic, anti-inflammatory, and anti-diarrheal properties of plant-derived charcoal [35]. One proposed mechanism is that charcoal adsorbs gastrointestinal moisture and adheres to the mucosal surface, aiding in the removal of inflamed tissue and supporting ulcer healing [35]. Intestinal inflammation is also known to disrupt the expression and organization of tight junction proteins, weakening barrier integrity [36]. Tight junction components, such as *Claudin-3* and *ZO-1*, play essential roles in preventing the translocation of harmful luminal substances, bacteria, and toxins into the systemic circulation [37,38]. In this study, *Claudin-3* expression increased with rising biochar inclusion and peaked in the G3 group before declining at higher levels. This trend closely mirrored the expression patterns of the inflammatory markers, suggesting that dietary biochar modulates tight junction integrity in concert with its anti-inflammatory effects in *M. salmoides*.

The intestinal tract functions as a complex microecosystem in which host–microbiota interactions play essential roles in regulating inflammation, metabolic activity, and susceptibility to or disease resistance [39]. Alpha diversity within this ecosystem is commonly assessed using the Chao1 and Ace indices for species richness and the Shannon and Simpson indices for species diversity [40]. Previous studies have shown varied responses of intestinal microbiota diversity to biochar: algal biochar supplementation in tilapia had no significant effect on Chao1, Shannon, or Simpson indices [41]; bamboo charcoal powder increased ACE, Shannon, and Chao1 indices in loach (*Paramisgurnus dabryanus*) [22], whereas biochar exposure in African catfish (*Ictalurus punctatus*) following nanoparticle and microplastic exposure decreased Chao1, Shannon, and ACE indices [42]. In this study, all α-diversity indices, except Simpson’s index, declined following biochar supplementation. This reduction may be associated with the increased abundance of *Lactococcus* and Firmicutes. *Lactococcus* contributes to intestinal digestion and absorption, glucose metabolism, and immune regulation [43], whereas *Mycoplasma* and *Klebsiella* are opportunistic pathogens that can disrupt intestinal homeostasis and induce inflammation when overgrown [44]. Reggi et al. [45] also reported that bioactive compounds in biochar can inhibit the synthesis of quorum-sensing signals, disrupt biofilm formation, and impair flagellar motility, thus reducing bacterial pathogenicity.

At the phylum level, the G3 group showed increased Firmicutes and decreased Proteobacteria and Fusobacteriota relative to the control group, patterns consistent with observations in tilapia [41] and loach [22]. These results demonstrate that biochar exerts selective inhibitory effects on pathogenic bacteria and may enhance the intestinal environment by influencing carbohydrate and lipid metabolism.

Metabolomic analysis was performed to elucidate further how dietary biochar supplementation regulates intestinal metabolism in *M. salmoides*. The results indicated significant enrichment of pathways associated with NET formation and isoflavonoid biosynthesis following biochar intake. Various flavonoid metabolites, including baicalin, genistein, and apigenin, were significantly upregulated. Previous research has shown that baicalin inhibits the *NF-κB* signaling pathway, reducing pro-inflammatory mediators such as *TNF-α* and *IL-6*, which may also contribute to decreased NET formation [46]. Genistein has been reported to downregulate CIP2A, induce cell cycle arrest, and exert potent anticancer activity [47]. Thus, the elevated levels of baicalin and genistein observed in this study likely reflect attenuated inflammation and enhanced immune regulation, consistent with the reduced expression of inflammatory genes in our results. This aligns with findings by Lin et al. [48], who demonstrated that biochar alleviated immunosuppression in zebrafish (*Danio rerio*) exposed to MC-LR by activating the Toll-like receptor/MyD88 pathway.

Apigenin is known to induce cell cycle arrest and apoptosis and has also been associated with reduced anxiety and depressive behaviors, which influence feeding activity [49]. Moreover, apigenin, genistein, and baicalin all activate the Nrf2 pathway, enhancing cellular antioxidant defenses and mitigating oxidative stress [50]. These findings suggest that dietary biochar may promote fish vitality through immunomodulatory, antioxidant, and anti-inflammatory mechanisms, potentially mediated by NF-κB inhibition, Nrf2 activation, and mitigation of stress-related responses.

Biochar supplementation significantly increased intestinal levels of taurochenodeoxycholate-3-sulfate, taurochenodeoxycholate-7-sulfate, taurocholate 3-sulfate, and arginylmethionine. Sulfated taurocholic acids are modified bile acids that help regulate bile acid synthesis, selectively inhibit pathogen colonization, and reduce intestinal permeability [51]. Arginylmethionine plays a vital role in epithelial cell proliferation, protein synthesis, and immune regulation. Han et al. [52] similarly found that biochar elevated intestinal chenodeoxycholate and bile acid levels in mice, reduced the enteritis marker tetradecanoic acid, and was accompanied by increased abundances of *Firmicutes* and *Bacteroidetes*. In broilers, a complex of activated charcoal and herbal extracts upregulated CYP7A1 expression, enhanced bile acid synthesis, and mitigated mycotoxin-induced inflammation and oxidative stress [53]. Therefore, the elevated levels of taurocholic acid sulfates and arginylmethionine observed in this study likely reflect increased bile acid metabolism and improved gut microbial composition, contributing to a stronger intestinal barrier. This interpretation aligns with the increased prevalence of beneficial taxa, such as Firmicutes and *Lactococcus*, and with the upregulation of tight junction-related genes.

Soyasaponin and etoxazole levels were significantly reduced following biochar supplementation. Soyasaponin is known to damage fish respiratory tissues and induce intestinal inflammation [54], whereas etoxazole interferes with thyroid hormone pathways, impairing growth and development [55]. Their decline may be attributed to their physicochemical properties: soyasaponin’s lipophilicity and etoxazole’s strong hydrophobicity facilitate their adsorption onto the aromatic ring structures of biochar [56,57]. Spearman correlation analysis supported this interpretation, revealing strong negative associations between soyasaponin and etoxazole and the abundance of *Lactococcus*, as well as negative correlations between apigenin, arginylmethionine, and genistein and *Klebsiella*. These findings provide additional insight into the protective effects of biochar on intestinal health in *M. salmoides*.

Although this study was limited to a 56-day trial without lower-dose treatments, serum indicators, or functional gene validation, and utilized biochar derived from a single feedstock (*Eucalyptus*), it offers the first comprehensive evidence that dietary biochar can beneficially modulate the intestinal environment of largemouth bass. Future work extending the trial duration, evaluating multiple biochar feedstocks, and integrating multi-omics approaches will help refine dose–response relationships and support the development of practical, biochar-based feed strategies for sustainable aquaculture.

## 5. Conclusions

Dietary supplementation with 10 g kg^−1^ of Eucalyptus branch-derived biochar increased the intestinal anti-inflammatory response, strengthened gut barrier integrity, and improved the microbial composition of largemouth bass (*Micropterus salmoides*). These benefits appear to be linked to alterations in bile acid metabolism and isoflavonoid biosynthesis. Through a multi-omics framework, this study clarifies the protective mechanisms by which biochar supports intestinal health in *M. salmoides*, offering valuable evidence for the potential incorporation of Eucalyptus-derived biochar into aquafeeds.

## Figures and Tables

**Figure 1 biology-14-01754-f001:**
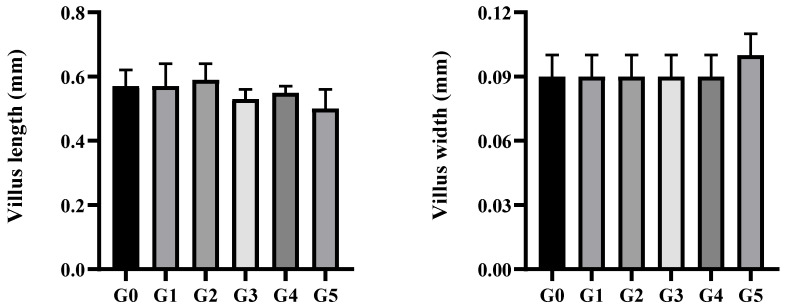
Intestinal structure of *Micropterus salmoides* fed diets supplemented with 0 (G0), 2.5 (G1), 5.0 (G2), 10.0 (G3), 20.0 (G4), and 40.0 (G5) g kg^−1^ *Eucalyptus* biochar (mean ± SE, *n* = 9). Bars with different letters represent significant differences (*p* < 0.05).

**Figure 2 biology-14-01754-f002:**
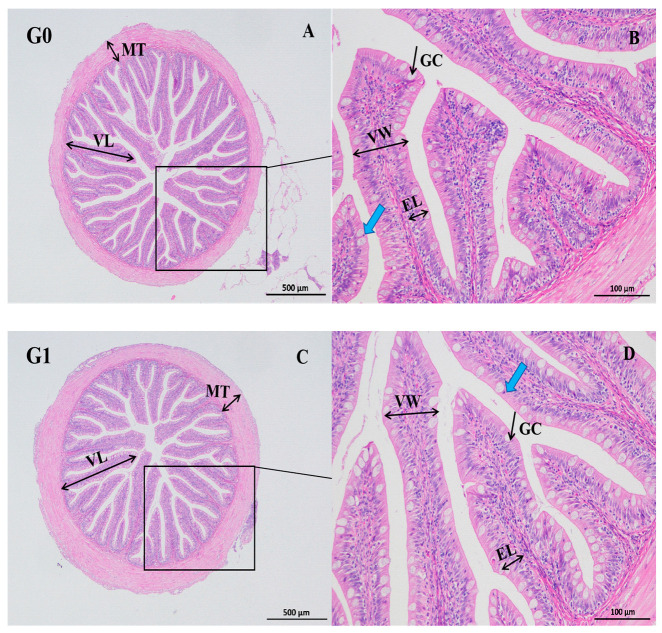
Intestinal histomorphology of *Micropterus salmoides* fed diets containing 0 (G0: (**A**,**B**)), 2.5 (G1: (**C**,**D**)), 5.0 (G2: (**E**,**F**)), 10.0 (G3: (**G**,**H**)), 20.0 (G4: (**I**,**J**)), and 40.0 (G5: (**K**,**L**)) g kg^−1^ *Eucalyptus* biochar. VL, villus length; MT, muscle layer thickness; VW, villus width; GC, goblet cells; EL, epithelial length. Scale bars: 500 μm (**A**,**C**,**E**,**G**,**I**,**K**) and 100 μm (**B**,**D**,**F**,**H**,**J**,**L**); blue arrow, lymphocyte infiltration.

**Figure 3 biology-14-01754-f003:**
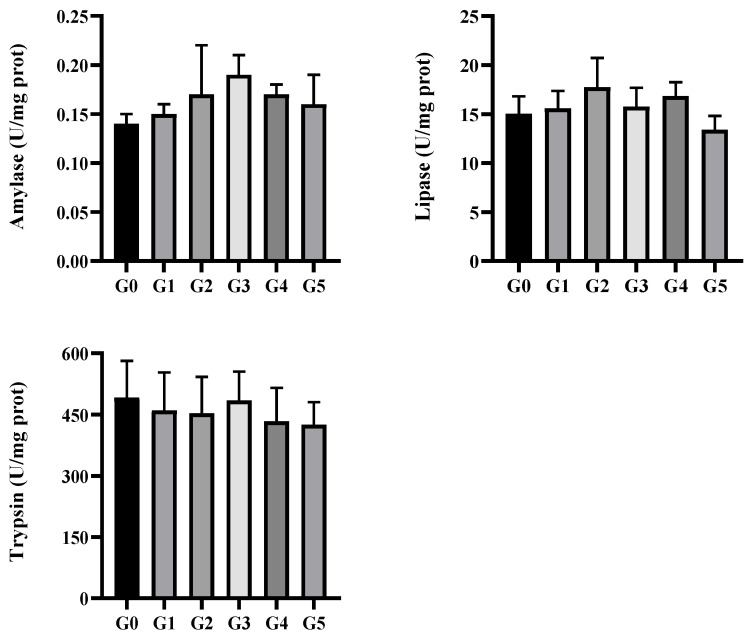
Intestinal digestive enzymes of *Micropterus salmoides* fed diets containing 0 (G0), 2.5 (G1), 5.0 (G2), 10.0 (G3), 20.0 (G4), and 40.0 (G5) g kg^−1^ *Eucalyptus* biochar (mean ± SE, *n* = 9).

**Figure 4 biology-14-01754-f004:**
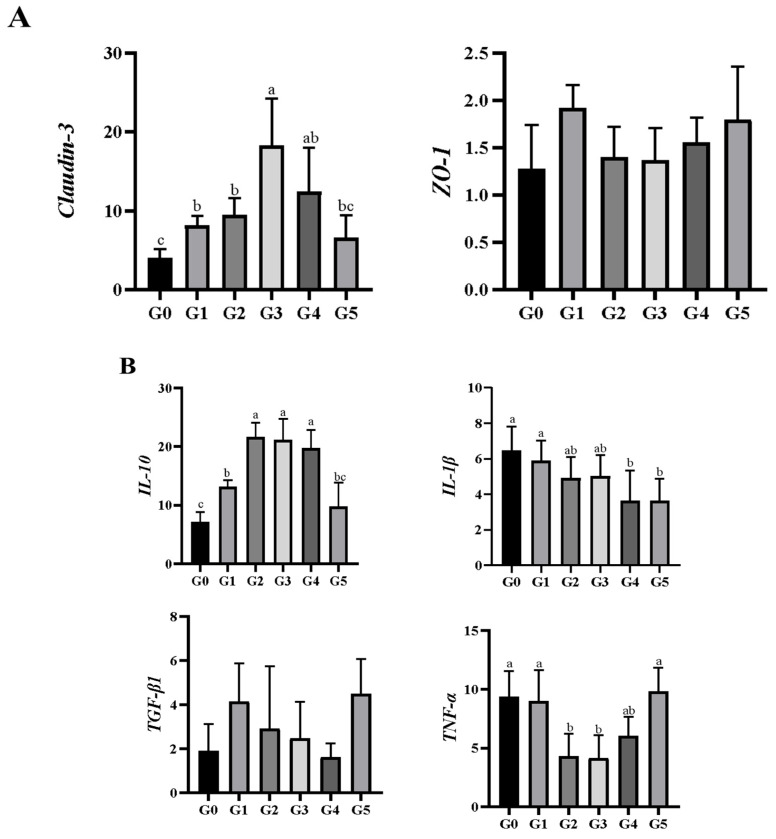
Intestinal gene expression levels of (**A**) tight junction-related and (**B**) inflammation-related markers in *Micropterus salmoides* fed diets containing 0 (G0), 2.5 (G1), 5.0 (G2), 10.0 (G3), 20.0 (G4), and 40.0 (G5) g kg^−1^ *Eucalyptus* biochar (mean ± SE, *n* = 9). Bars with different letters represent significant differences (*p* < 0.05).

**Figure 5 biology-14-01754-f005:**
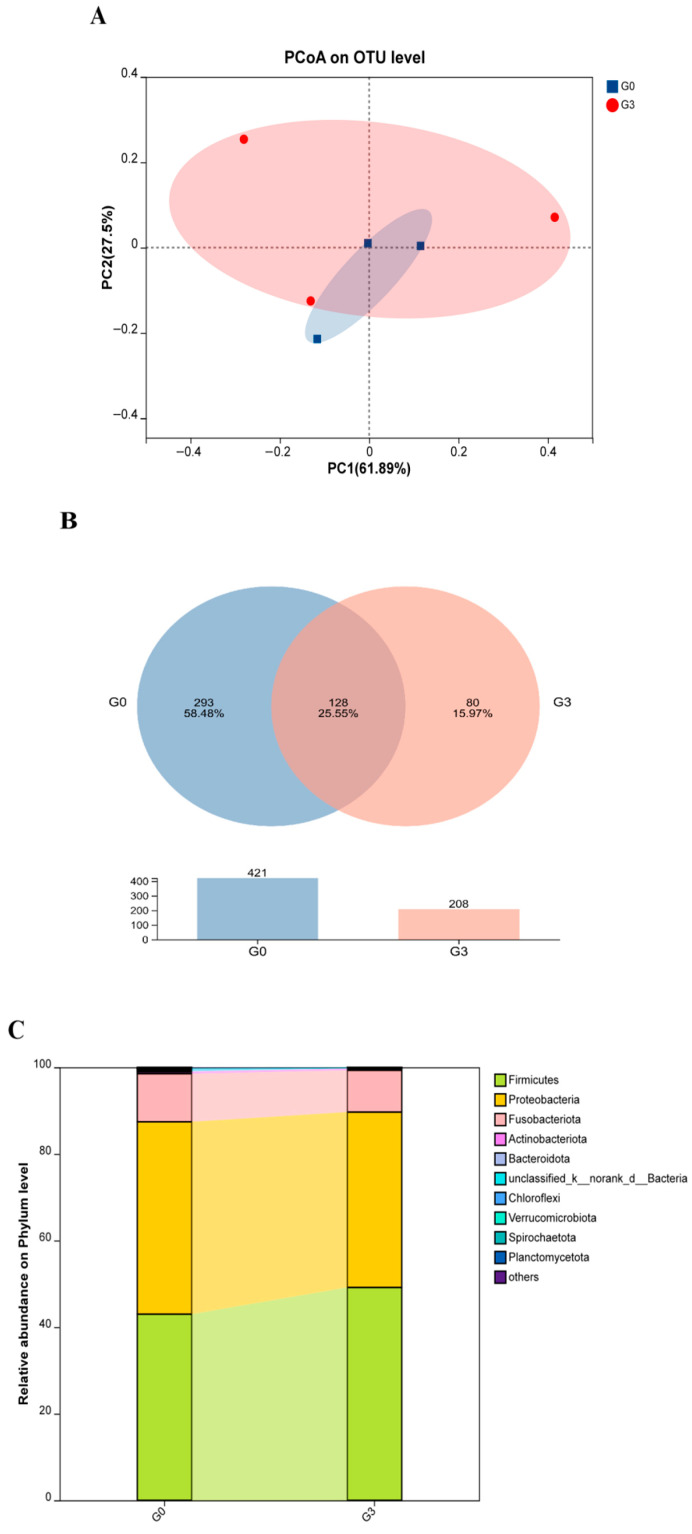
Intestinal microbial compositions of *Micropterus salmoides* fed 0 (G0) or 10.0 (G3) g kg^−1^ *Eucalyptus* biochar: (**A**) Principal component analysis (PCA) plot. (**B**) Venn diagram. (**C**,**E**) Gut microbiota composition at the phylum level. (**D**,**F**) Gut microbiota composition at the genus level (*n* = 3).

**Figure 6 biology-14-01754-f006:**
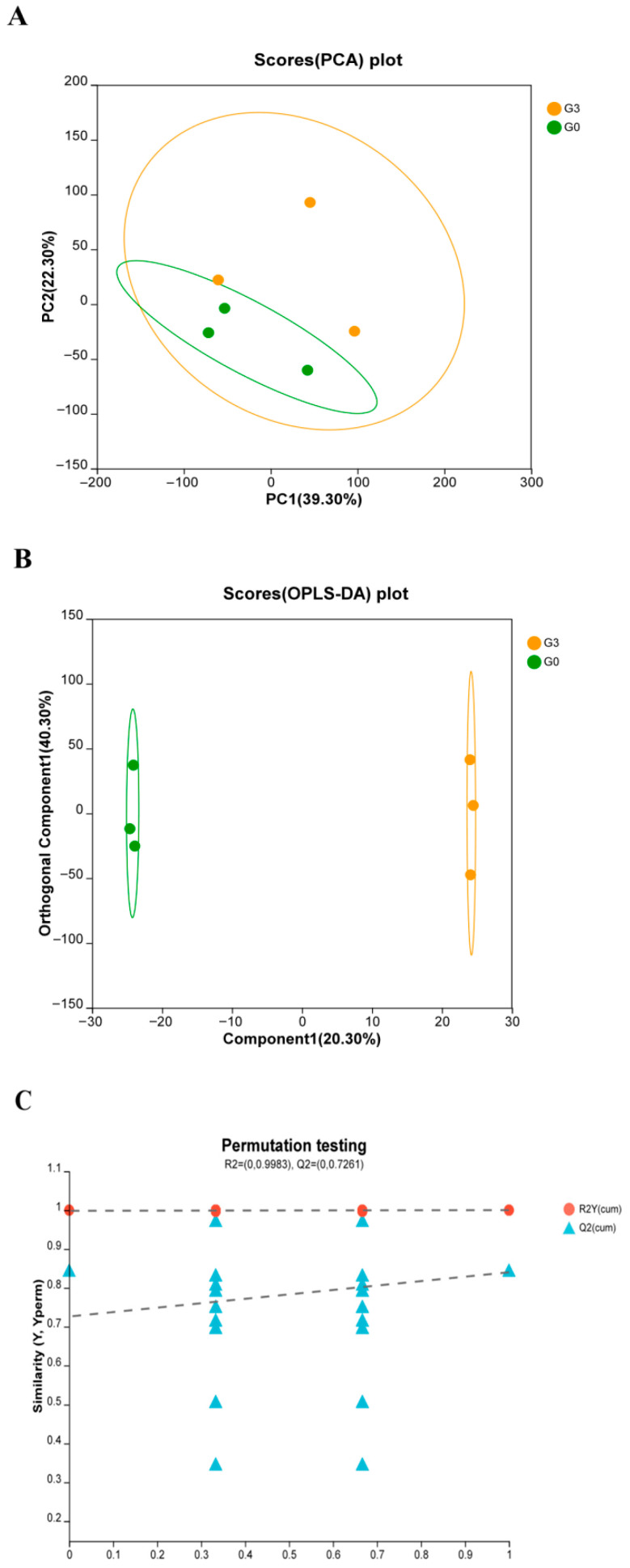
Metabolomics discrimination of *Micropterus salmoides* fed 0 (G0) or 10.0 (G3) g kg^−1^ *Eucalyptus* biochar: (**A**) PCA model. (**B**) OPLS-DA model. (**C**) Permutation test of the OPLS-DA model (*n* = 3).

**Figure 7 biology-14-01754-f007:**
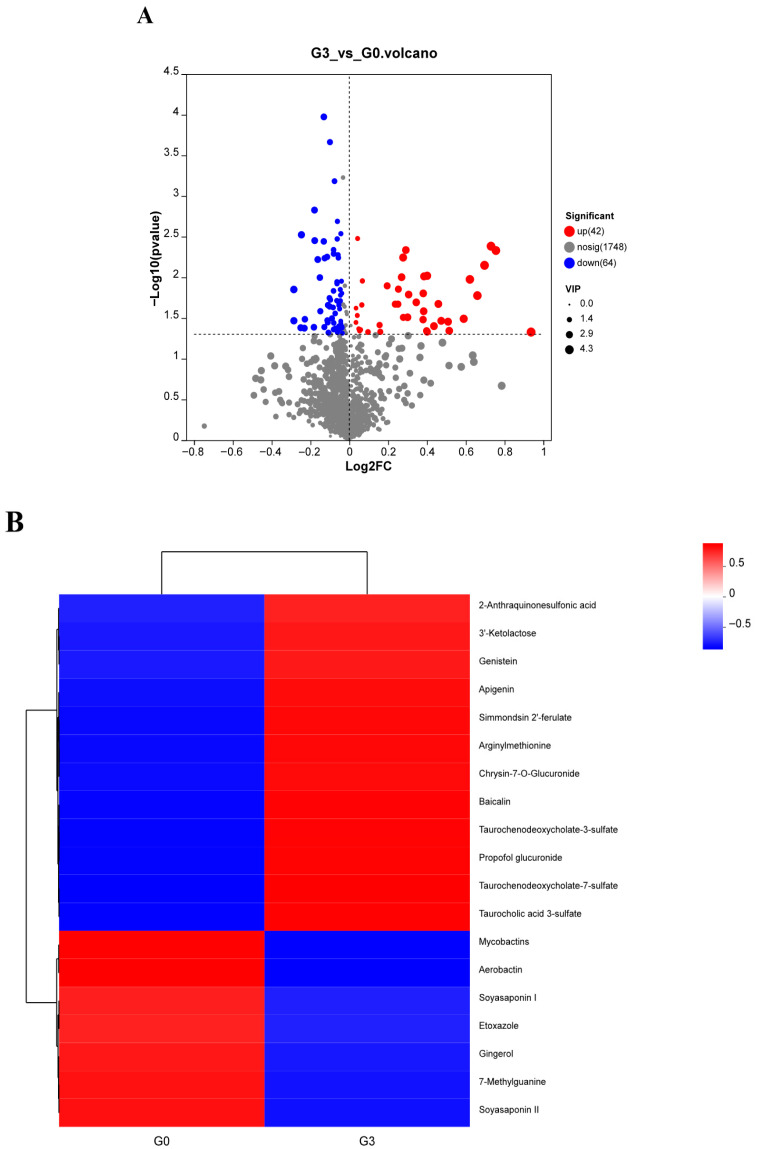
(**A**) Volcano plot and (**B**) cluster heatmap of differential metabolites in *Micropterus salmoides* fed 0 (G0) or 10.0 (G3) g kg^−1^ *Eucalyptus* biochar.

**Figure 8 biology-14-01754-f008:**
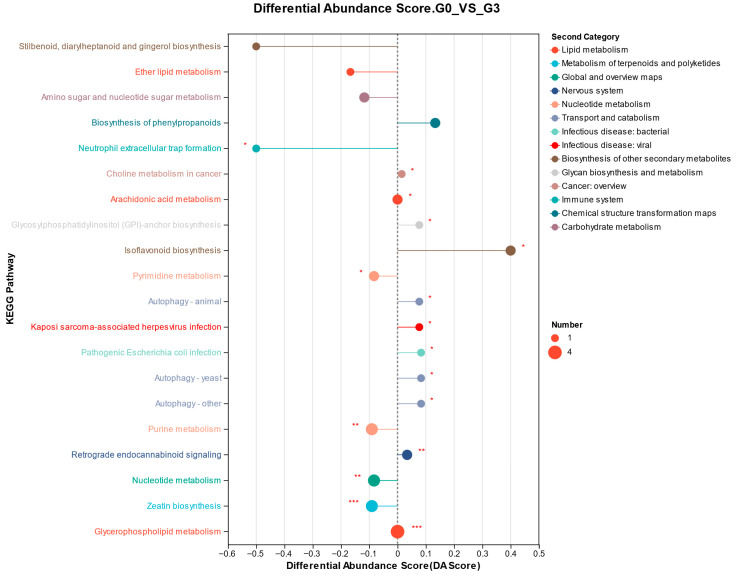
KEGG pathway enrichment bubble plot showing metabolic pathway differences in *Micropterus salmoides* fed 0 (G0) or 10.0 (G3) g kg^−1^
*Eucalyptus* biochar. * indicates *p*-value < 0.05, ** indicates *p*-value < 0.01, *** indicates *p*-value < 0.001. The length of the line segment represents the absolute value of the differential abundance score. The longer the line segment on the right side of the central axis, the more the overall expression of the pathway tends to be upregulated.

**Figure 9 biology-14-01754-f009:**
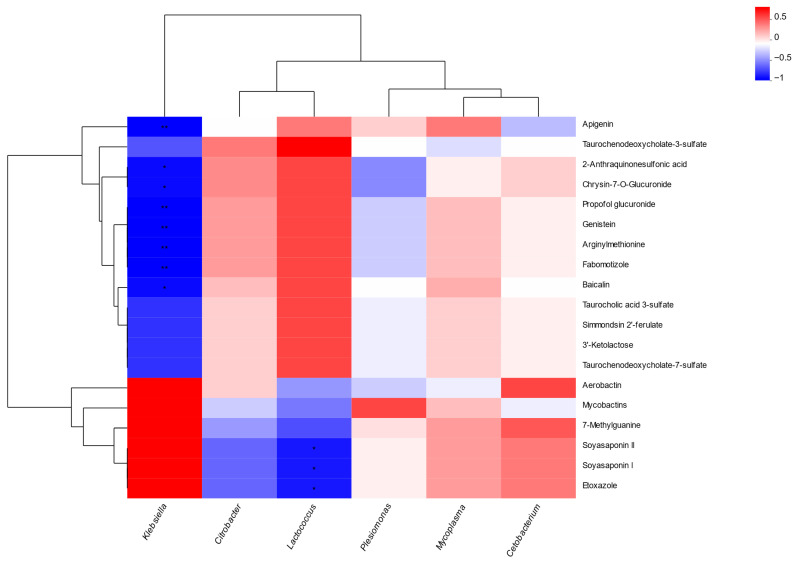
Correlation heatmap of metabolites and microbial taxa in *Micropterus salmoides* fed 0 (G0) or 10.0 (G3) g kg^−1^ *Eucalyptus* biochar. * indicates *p*-value < 0.05, ** indicates *p*-value < 0.01.

**Table 1 biology-14-01754-t001:** Ingredients and proximate composition (g kg^−1^ DM) of the experimental diets.

Items	Group					
	G0	G1	G2	G3	G4	G5
Ingredients						
Fish meal	450.0	450.0	450.0	450.0	450.0	450.0
Soybean meal	160.0	160.0	160.0	160.0	160.0	160.0
Soybean protein concentrate	166.0	166.0	166.0	166.0	166.0	166.0
α-starch	85.0	85.0	85.0	85.0	85.0	85.0
Fish oil	20.0	20.0	20.0	20.0	20.0	20.0
Soybean oil	35.0	35.0	35.0	35.0	35.0	35.0
Soybean lecithin	10.0	10.0	10.0	10.0	10.0	10.0
Vitamin C phosphate ester	3.0	3.0	3.0	3.0	3.0	3.0
Vitamin premix ^1^	1.0	1.0	1.0	1.0	1.0	1.0
Mineral premix ^2^	5.0	5.0	5.0	5.0	5.0	5.0
Ca(H_2_PO_4_)_2_	15.0	15.0	15.0	15.0	15.0	15.0
Choline chloride	5.0	5.0	5.0	5.0	5.0	5.0
Betaine	5.0	5.0	5.0	5.0	5.0	5.0
Cellulose	40.0	37.5	35.0	30.0	20.0	0.0
*Eucalyptus* biochar	0.0	2.5	5.0	10.0	20.0	40.0
Nutrient levels						
Crude protein	472.7	495.1	491.3	479.8	484.3	473.6
Crude lipid	76.6	79.1	78.9	82.8	87.7	87.8
Moisture	83.4	53.7	62.7	62.5	75.0	84.7
Ash	110.8	113.4	111.4	114.0	116.5	116.9
Ca	17.8	18.0	18.0	18.2	19.0	20.9
Total phosphorus	16.3	16.5	16.4	16.4	17.4	16.9

^1^ Each kilogram of the vitamin premix contained vitamin E, 50 IU; vitamin B_1_, 1 mg; choline, 1000 mg; vitamin B_6_, 5 mg; vitamin K_3_, 1 mg; nicotinic acid, 10 mg; vitamin D_3_, 2000 IU; biotin, 0.14 mg; vitamin B_2_, 6 mg; D-calcium pantothenate, 20 mg; vitamin A, 2500 IU; folic acid, 1 mg; vitamin C, 50 mg. ^2^ Each kilogram of the mineral premix contained CuSO_4_·H_2_O, 7 mg; NaCl, 1200 mg; KI, 8 mg; FeSO_4_·H2O, 13 mg; MnSO_4_·H_2_O, 32 mg; ZnSO_4_·H_2_O, 60 mg.

**Table 2 biology-14-01754-t002:** Primer sequences used for qPCR analysis.

Gene	Forward Sequence (5′-3′)	Genbank No.	Length (bp)	Tm (°C)
*Claudin-3*	Forward: CATCCTTGCTGGCCTTTTGG	XM_038708625.1	128	58.9
	Reverse: AGCCAATGTAGAGCGATGC			
*IL-10*	Forward: CGGCACAGAAATCCCAGAGC	XM_038696252.1	119	60.9
	Reverse: CAGCAGGCTCACAAAATAAACATCT			
*IL-1β*	Forward: ACATGACGGAAGTTCAGGAT	XM_038733429.1	150	57.3
	Reverse: GCTGCCTGCTATAGTTGGTT			
*TGF-β1*	Forward: GCTCAAAGAGAGCGAGGATG	XM_038693206.1	118	57.8
	Reverse: TCCTCTACCATTCGCAATCC			
*TNF-α*	Forward: CTTCGTCTACAGCCAGGCATCG	XM_038710731.1	161	63.1
	Reverse: TTTGGCACACCGACCTCACC			
*ZO-1*	Forward: GCTTACCTCACTGTGCGTCT	XM_038701018.1	232	57.2
	Reverse: GCATCCTCTTCATTTTATCCC			
*β-actin*	Forward: GGTGTGATGGTTGGTATGG	XM_038712920.1	156	55.2
	Reverse: CTCGTTGTAGAAGGTGTGAT			

*IL-10*, interleukin-10; *IL-1β*, interleukin-1β; *TGF-β1*, transforming growth factor-β1; *TNF-α*, tumor necrosis factor-α; *ZO-1*, zonula occludens-1.

**Table 3 biology-14-01754-t003:** Intestinal microbiota α-diversity indices of *Micropterus salmoides*.

Items	Groups	
	G0	G3
Sobs	171.33 ± 80.68	93.00 ± 28.29
Coverage (%)	99.93 ± 0.02	99.94 ± 0.02
ACE index	201.32 ± 82.60	138.45 ± 49.54
Chao1 index	195.25 ± 83.04	117.97 ± 39.53
Shannon index	1.62 ± 0.11	1.29 ± 0.15
Simpson index	0.29 ± 0.03	0.38 ± 0.07

## Data Availability

The original contributions presented in this study are included in the article/Appendix A. Further inquiries can be directed to the corresponding authors.

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
