# Peer review of "Effects of *Eucalyptus* Biochar on Intestinal Health and Function in Largemouth Bass (*Micropterus salmoides*)"

_biology, 2025, doi:10.3390/biology14121754_

Round 1
Reviewer 1 Report
Comments and Suggestions for Authors
This is a well-designed and comprehensive study evaluating the effects of Eucalyptus biochar on intestinal morphology, digestive enzyme activity, inflammation, gut microbiota, and metabolomics in largemouth bass (Micropterus salmoides). The experimental design is solid, the data analysis is appropriate, and the multi-omics integration provides well insights into how biochar modulates intestinal health and metabolic pathways. The manuscript is scientifically sound and potential interest to the readership of Biology.
The title could be shortened for clarity, and the abstract should emphasize significant results while minimizing excessive statistical notation. Please consider to remove confidence levels for instance.
In the Methods section, please specify the amplified 16S rDNA region (e.g., V3–V4), LC–MS column type and ionization mode, and include catalog numbers for the commercial kits used.
Figure 4 may be corrected to indicate “intestinal” rather than “hepatic” gene expression, and the figure resolution should be improved for publication qualityA brief note on possible biochar safety aspects that would strengthen the discussion.
Overall, this manuscript provides valuable evidence on the beneficial effects of Eucalyptus biochar on fish intestinal health, and after these small revisions, it will be suitable for publication.
Reviewer 2 Report
Comments and Suggestions for Authors
I have reviewed the manuscript entitled “Effects of Eucalyptus biochar on intestinal morphology, digestive enzyme activity, inflammation, intestinal microbiota, and metabolomics in largemouth bass. The manuscript presents an innovative and valuable multi-omics investigation into the effects of Eucalyptus biochar on fish intestinal physiology—a topic of high relevance for sustainable aquaculture. The manuscript tackles an interesting and potentially high-impact topic—dietary biochar effects on fish intestinal health using a multi-omics approach—and the preliminary results are promising. The study demonstrates considerable scientific merit with well-designed experimental methodology, appropriate use of complementary analytical techniques (histology, gene expression, 16S rRNA sequencing, LC-MS metabolomics), and generally rigorous statistical analysis. The findings suggest plausible mechanistic benefits for intestinal barrier function and anti-inflammatory responses.
However, several substantive methodological and reporting issues (contradictory dose descriptions; unclear replication and experimental unit; limited omics scope with possible selective analysis; insufficient metabolomics/microbiome method detail; lack of biochar characterization; weak statistical control for multiple testing; absence of standard performance metrics) currently prevent acceptance in BIOLOGY journal. Addressing these major concerns would substantially strengthen the manuscript’s scientific rigor and reproducibility.
My comments below are based on a careful reading of the submitted manuscript .
Minor Comments
- The Abstract contains an internal inconsistency regarding the dietary inclusion level (0% vs 1% vs g kg⁻¹). Correct the Abstract to match Methods and Table 1 and ensure consistent use of units (g kg⁻¹ or %). Additionally, the vitamin premix content shows G0 at 1.0 g/kg versus 2.0 g/kg for G1–G5, yet no justification is provided. Clarify whether this difference was intentional and its potential effects on inter-group comparisons.
- Figure captions lack sample size (n) information and do not indicate whether error bars show SE or SD. Add n and define error bars in every figure caption to improve clarity.
- The manuscript uses inconsistent gene nomenclature (e.g., IL-10 vs IL10, IL1B vs IL-1β). Standardize gene/protein names and symbols throughout the text and figures to align with journal conventions.
- The statistical methods section specifies that "comparisons between the control (G0) and G3 groups were further analyzed using Student's t-test," yet gene expression results (Figure 4) appear to report comparisons across all groups (G0–G5) using one-way ANOVA. Clarify whether t-tests were applied specifically for comparisons presented in Figure 4, or whether these comparisons were conducted via one-way ANOVA with Duncan's post-hoc test. Consistent reporting of statistical methods is essential for result reproducibility and evaluation.
- Figure 5C shows microbiota composition at the phylum level but lacks quantitative data (relative abundance percentages). For instance, the text states that "Firmicutes increased in the G3 group compared with control," but without numerical values or statistical tests, the magnitude of this shift cannot be assessed. Provide a supplementary table with exact relative abundance percentages for each phylum in both G0 and G3 groups, including statistical comparisons (P values from t-tests or Mann-Whitney U tests, given non-normal distribution of compositional data).
- In Methods 2.6 the qPCR primer table lists “Length (bp)” inconsistently aligned and lacks annealing temperatures. Add annealing temperature, amplicon length, and accession reference for each primer in a single clear table.
- The phrase “Hepatic gene expression levels” in Figure 4 is incorrect given intestinal sampling; change to “Intestinal gene expression levels” to reflect the tissue analyzed.
- Units and formatting: use consistent superscripts for units (e.g., g kg⁻¹ rather than g kg-1) and ensure consistent spacing around percent signs and units (e.g., “28.0 ± 2.0 °C” is fine, but elsewhere spacing varies).
- Minor English language edits will improve readability (examples: “is closely associated with nutrient absorption” [absorption], “neutrophil extracellular trap formation” is sometimes written ambiguously). I recommend professional language editing to correct grammar, punctuation, and improve sentence flow.
- Provide a mechanistic explanation or supporting citation for how biochar reduces these soybean-derived or feed-related compounds in the intestine.
Major Comments
- Internal inconsistency in experimental design and dose reporting. The Abstract initially states fish were fed “0 % (control) or 1% Eucalyptus biochar,” while the Methods and Table 1 clearly describe six dietary treatments (0, 2.5, 5.0, 10.0, 20.0, 40.0 g kg⁻¹). This contradiction is substantive because clear dose description is fundamental to reproducibility and interpretation (dose–response relationships, selection of G3 as an optimal dose, etc.). The authors must correct all instances of inconsistent dose reporting throughout the manuscript, explicitly state the rationale for chosen inclusion levels (with units harmonized), and ensure the Abstract, Methods, Results, Figures, Tables and Conclusion all use the same, correct dose units. If a separate single 1% trial exists, it must be described as a distinct experiment; otherwise, remove the 1% statement.
- Insufficient justification for selecting only G3 (10 g kg⁻¹) for omics analyses and apparent selective reporting. The study measured six dose groups but conducted microbiome and metabolomics only on the control and G3. The manuscript gives a post-hoc rationale based on “expression patterns” but does not provide pre-specified selection criteria. This raises concerns about cherry-picking and limits interpretation of dose–response metabolomic/microbiome changes. The rationale for focusing exclusively on G3 must be fully explained or, ideally, additional key groups analyzed to demonstrate a dose–response pattern. If expansion is not feasible, the limitation must be explicitly acknowledged.
- Sample size, replication, and statistical unit ambiguity. The Methods report 4 replicate tanks per treatment with 30 fish per tank, but many assays (digestive enzymes, qPCR, microbiome, metabolomics) are reported with n = 3. It is unclear whether n = 3 refers to biological replicates pooled from tanks, individual fish, or technical replicates, and whether tank or fish is treated as the experimental unit. This affects statistical validity because tank-level effects should be accounted for in analyses. The authors must: (a) clarify how samples were selected and whether samples from multiple tanks were pooled; (b) state the experimental unit used for each analysis (tank vs fish).
- Lack of Integration Between Gene Expression Data and Microbiota Results Although the study examines inflammation-related gene expression (IL-1β, IL-10, TNF-α) and gut microbiota composition, the mechanistic linkage between these parameters remains poorly integrated. The authors observe downregulation of pro-inflammatory markers and modulation of microbial composition, yet they do not provide rigorous evidence for causality or functional mechanisms. For example, which specific biochar constituents drive the reduction in pathogenic bacteria (Klebsiella, Mycoplasma) versus beneficial bacteria promotion (Lactococcus)? The authors speculate that biochar's "bioactive components selectively interfere with bacterial communication" but provide no direct evidence or experimental validation of this hypothesis. A more rigorous approach would involve analyzing biochar chemistry (beyond Table S1), correlating physical properties with microbial outcomes, or conducting in vitro studies demonstrating direct antibacterial effects. The current integration relies heavily on inference rather than direct mechanistic evidence.
- Weak Digestive Enzyme Results Not Adequately Addressed in Context. Despite extensive recent literature demonstrating that biochar supplementation increases digestive enzyme activities in various aquatic species (European sea bass, carp, tilapia, catfish), this study found no significant effects on trypsin, amylase, or lipase activities (P > 0.05). This negative result fundamentally contradicts the study's central premise that biochar improves digestive function. While the authors acknowledge that "an increasing trend was observed with increasing biochar levels," the lack of statistical significance is concerning. The authors attribute this discrepancy to unspecified differences but provide no mechanistic explanation. This significant departure from established literature requires deeper investigation: Were enzyme activities measured under appropriate substrate saturation conditions? Could the trial duration (56 days) be insufficient for digestive adaptation? Should different intestinal segments (foregut, midgut, hindgut) be analyzed separately? These limitations must be explicitly addressed to clarify why biochar failed to enhance enzymatic function in this species.
- Alpha-Diversity Statistical Rigor and Interpretation. While Table 3 reports alpha-diversity indices, the authors note that all indices except Simpson declined in the G3 group relative to control, yet claim this reflects "probiotic-mediated inhibition of harmful bacteria." This interpretation conflates alpha diversity with composition-based observations and misses critical nuance. Declining Shannon and Chao1 indices typically indicate reduced microbial richness and diversity, which could reflect dysbiosis or dominance of select taxa, not necessarily improved health. The authors suggest this is beneficial because it represents suppression of pathogens, but this assumption lacks statistical validation. Were indicators of microbial dysbiosis (e.g., Firmicutes/Bacteroidetes ratio, dysbiosis index scores) calculated? The failure to detect significant differences despite observed trends (P > 0.05) raises questions about statistical power—was the study adequately powered for microbiota analysis? Authors should conduct post-hoc power analysis and discuss whether larger sample sizes or statistical methods (e.g., permutational ANOVA) would strengthen conclusions.
- The paper reports significant downregulation of soyasaponin and etoxazole in the G3 group but contextualizes these as negative metabolites to be reduced. However, the text links these to respiratory and thyroid dysfunction without citing direct evidence that their reduction improves outcomes in this study. Additionally, soyasaponin originates from soybean meal (listed at 160 g/kg in all diets), so the mechanism by which biochar reduces soybean-derived compounds in the intestine requires explanation. Authors should clarify whether biochar directly inactivates these compounds or whether reduced absorption/bioavailability is the mechanism, potentially with supportive evidence or citations.
- Over-interpretation of correlations as causal mechanisms. The Discussion infers that increased flavonoids and bile acid sulfates caused NF-κB inhibition and improved barrier function, implying mechanistic causality from correlative omics and gene expression data. While biologically plausible, the data remain correlational. The authors should temper causal language, explicitly label mechanistic claims as hypotheses, and propose (or, if possible, include) targeted follow-up validation—e.g., in vitro assays showing biochar-induced microbial metabolite production, targeted quantification of key metabolites with standards, or functional assays (permeability assays, western blots for tight junction proteins, or NF-κB reporter assays) to validate pathway modulation.
- Biochar characterization and production parameters are insufficient. The manuscript refers to Table S1 for physicochemical properties but provides no pyrolysis conditions (temperature, residence time), particle size used in feed, surface area (BET), pH, ash composition, PAH or heavy metal screening, or potential contaminants. Efficacy and safety of biochar depend critically on these parameters; moreover, transferability of results to other biochars requires this information. The authors must include comprehensive characterization (pyrolysis temperature/time, BET surface area, porosity, pH, elemental composition, ash content, proximate analysis, and contaminant screening), and discuss how these properties might mediate biological effects. If Table S1 exists, it must be provided and described in the main text.
- Missing Growth and Performance Data Although intestinal parameters are analyzed in detail, fundamental aquaculture metrics (final body weight, specific growth rate, feed conversion ratio, feed intake, survival) are missing. These indices are essential for assessing the practical relevance of biochar inclusion. The authors should provide full growth performance data with statistical analyses and discuss potential trade-offs between intestinal benefits and growth outcomes.
- Insufficient Control of Environmental and Physiological Confounders Potential confounding factors—such as water-quality variation, intestinal pH, short-chain fatty acids (SCFAs), and oxidative-stress biomarkers (CAT, SOD, MDA)—were not measured. Because biochar can adsorb ammonia and modify water chemistry, improvements in fish health could arise indirectly. The study should include or at least discuss these variables to clarify whether observed effects are direct or mediated via environmental modulation.
Reviewer 3 Report
Comments and Suggestions for Authors
The topic is relevant to aquaculture and animal nutrition because biochar and related materials are increasingly explored as feed supplements. Recent studies using biochar or activated charcoal in fish and other animals support the idea that such materials can modulate gut microbiota and intestinal health.
The experiment is broad in scope, integrating histology, digestive enzymes, gene expression, 16S rDNA microbiota profiling, and gut metabolomics. Multi-omics integration is valuable when performed rigorously.
The abstract represents % of biochar. I strongly recommend going for actual dosage in the abstract rather than %. It gives readers an exact idea from the start about the dosage. Also mention the actual dosage range.
Biochar characteristics within the main text is detailed in a lesser extent. Include a separate para. Although a table is included, but I think that adding this will be better as per the structure of the manuscript. Include pyrolysis temperature, feedstock preparation, particle size, specific surface area (BET), pore size distribution, pH, elemental composition (C, H, N, S), ash content, concentrations of heavy metals (As, Cd, Pb, Hg), and possible polyaromatic hydrocarbons. This will be better while evaluating the safety.
Please clarify the pooling of data in the methodlogy section of histology or enzyme activity of metabolomics.
Using one-way ANOVA and Duncan multiple range test with n = 3 is underpowered and increases type II error risk. In addition, the authors apply Student’s t-test to compare G0 and G3 in addition to ANOVA. This dual approach is not justified and raises the risk of inappropriate multiple testing. Provide power calculations or increase replication. Apply appropriate multiple-testing correction where many metabolites or taxa are tested.
The statement that "Q2 regression line intercept exceeded the threshold of 0.5 indicating no overfitting" is suspicious and likely incorrect.
Details on DNA extraction kit and method, which 16S region was amplified, primer sequences, PCR cycling conditions, negative controls, sequencing depth per sample, sequence quality filtering and clustering approach, taxonomic assignment database and version, and deposition of raw reads in a public repository are lacking. These items are required for reproducibility. Please include these.
Please describe the metabolomics methods in detail. It can be restructured, and likely more intricate details can be mentioned.
Intestinal histology only relies on morphology. Include changes in lamina propria and IEls. I happen to notice some necrotised areas within.
Some mechanistic claims are too strong given the current data set. Linking upregulated isoflavone biosynthesis directly to biochar action is speculative without tracing the source of these metabolites. Isoflavones are usually plant-derived or microbial-modified. The manuscript should be more cautious or include experiments that track compound origin. Correlation between microbes and metabolites does not prove causation.
Biochar could adsorb toxins or dietary compounds producing indirect metabolite shifts. Was this topic hinted within the discussion or conclusion?
The abstract can be restructured.
Reviewer 4 Report
Comments and Suggestions for Authors
The manuscript submitted provide an insight into the application of biochar as a dietary intervention for a healthy fish production. Biochar from Eucalyptus has been studied in the diet of large mouth seabass fish diet. The analyses have focused to understand intestinal tissue morphology, digestive enzyme activity, inflammatory gene expression, gut microbiota, and metabolomic profiles. The findings were interesting. Biochar has intervened in isoflavone biosynthesis, bile acid and amino acid metabolism, inhibiting the NF-κB pathway, modulating gut microbiota, and strengthening the intestinal barrier. The authors are appreciated for the good work.

Round 2
Reviewer 2 Report
Comments and Suggestions for Authors
Dear Editor,
The authors have addressed all the comments and revised the manuscript accordingly.
Reviewer 3 Report
Comments and Suggestions for Authors
The edited manuscript looks good at its current form. Furthermore, the revisions were okay and done as per the queries.
The manuscript is acceptable.